# Family Support as Smoking Prevention during Transition from Early to Late Adolescence: A Study in 42 Countries

**DOI:** 10.3390/ijerph182312739

**Published:** 2021-12-02

**Authors:** Apolinaras Zaborskis, Aistė Kavaliauskienė, Charli Eriksson, Ellen Klemera, Elitsa Dimitrova, Marina Melkumova, Daniela Husarova

**Affiliations:** 1Faculty of Public Health, Medical Academy, Lithuanian University of Health Sciences, LT-44307 Kaunas, Lithuania; 2Faculty of Odontology, Medical Academy, Lithuanian University of Health Sciences, LT-44307 Kaunas, Lithuania; aiste.kavaliauskiene@lsmuni.lt; 3Department of Public Health Sciences, Stockholm University, SE-10691 Stockholm, Sweden; charli.eriksson48@gmail.com; 4Centre for Health Services Studies, Division of Low, Society and Social Justice, University of Kent, Canterbury CT2 7NS, UK; E.Klemera@kent.ac.uk; 5Institute for Population and Human Studies, Bulgarian Academy of Sciences & Plovdiv University Paisii Hilendarski, 1000 Sofia, Bulgaria; elitsa_kdimitrova@yahoo.com; 6Arabkir Medical Centre-Institute of Child and Adolescent Health, National Institute of Health, Yerevan 0014, Armenia; mmelkumova@mail.ru; 7Department of Health Psychology and Methodology Research, Faculty of Medicine, Pavol Jozef Šafárik University in Košice, 04011 Kosice, Slovakia; Daniela.Brindova@gmail.com

**Keywords:** adolescents, parents, family, support, smoking, prevention, HBSC

## Abstract

Family support has a beneficial impact on protecting health-risk behaviour in adolescents. This study aimed to explore whether family support is associated with risk of smoking during transition from early (11 years) to late (15 years) adolescence across 42 countries. The data from the cross-national Health Behaviour in School-aged Children (HBSC) study in 2017/2018 were employed (N = 195,966). Family support was measured using the four-item Family dimension of the Multidimensional Scale of Perceived Social Support (sum score 20 or more was categorised as high family support). Smoking was defined as a reported cigarette smoking at least 1–2 days in the last 30 days. The association between smoking and family support was assessed using a prevalence ratio (PR) obtained from the multivariate Poisson regression. Over two thirds of adolescents reported high levels of support from their family. Family support was found to significantly decrease with age in most of the countries, with the boys reported high level of family support more often than girls. The adolescents who reported having low family support also were more likely to smoke compared to their peers who reported having high family support (PR = 1.81; 95% CI: 1.71–1.91 in boys, and PR = 2.19; 95% CI: 2.08–2.31 in girls). The countries with a stronger effect of family support in reducing smoking risk indicated lower rates of adolescent smoking as well as lower increases in the cigarette smoking prevalence during the age period from 11 to 15 years. This study reinforces the need for family support, which is an important asset helping adolescents to overcome the risk of smoking during their transition from early to late adolescence.

## 1. Introduction

Adolescence is marked as a period of rapid developmental changes and often perceived as a time of changing behaviour and health across the life course [1,2]. As a transition from the childhood to adulthood, adolescence is a time of opportunity and vulnerability to health risk behaviour with lifelong consequences for health and well-being [3,4]. During this period smoking is most commonly initiated and addiction is likely to happen [5]. Adolescent girls and boys who start smoking believe that they will be able to stop soon and easily, but the addictive nature of nicotine causes most of them to develop a dependence on this substance and to continue smoking for many years [6]. In addition to the problems of tobacco addiction, smoking in adolescence has immediate consequences on physical health [7], it is linked with depressive symptoms [8,9], suicidal ideation [10] and with other addictive behaviours such as intensive alcohol consumption [11], cannabis use, or gambling [12]. Faced with this reality, the prevention of smoking at this age should be considered of high priority [13]. 

Family support, which can be defined as positive parent–child interactions grounded in open communication and high parental sensitivity and responsibility to their child’s needs, has a beneficial impact on the psychological well-being of adolescents as well as on protecting against poor health outcomes and health-risk behaviour [14,15,16]. Love, support, trust, and optimism from their family make adolescents feel safe and secure, and are powerful weapons against peer pressure, life’s challenges, and disappointments [17]. However, the role of parents and parent–adolescent relations undergo a process of change through transitions. Even though family support decreases from early to late adolescence [18] (pp. 31–34), parents continue to play a fundamental role in adolescent development, socialization, health, and well-being, and this role may be as important as it is in the early developmental stages, even though it is different and less noticeable [19]. 

Several studies have found a positive relationship between perceived family support (especially parental support) and adolescents’ mental and physical health and prevalence of engagement in health-risk behaviours [19]. A high level of parental support is associated with better emotional well-being, fewer internalizing and externalizing problems [20,21,22], and better educational outcomes [23]. A supportive family environment can also play a crucial role in health promotion, for example, assisting healthy changes of lifestyle. Family support is a protective factor against health-risk behaviour in ordinary samples [21,24,25] and against maladjustment in at-risk populations [26]. High parental support buffers against the negative consequences of adverse life events and peer-victimisation, especially among young female adolescents [27]. 

Family environment and health behaviours during adolescence is one of the foci of the cross-national Health Behaviour in School-aged Children (HBSC) study [28]. The previous reports of the study [18,29] have highlighted that over two thirds of adolescents reported high levels of support from their family, but wide cross-national variations were observed. Across most countries, younger girls and boys were more likely to report high family support. Significant gender differences were observed in less than half of countries/regions, with boys more likely to report higher levels of family support in most of these. In more than half of countries/regions, adolescent boys and girls from more affluent families reported higher levels of family support [29]. However, despite the evidence of developmental changes in perceived family support there is still a paucity of research about the possible changes in the impact of family support on smoking prevention during adolescence period [18]. The association between family factors and adolescent smoking habits may vary depending on the social and cultural context, so it is important to examine the relationships between adolescents from different countries. The HBSC study involves a wide network of researchers from more than 50 countries and regions, so its data allows us to successfully address such challenges [28]. 

The present article is aimed to contribute more specifically to the current evidence based on the role of family support during adolescent transitions. Consistent with recent research [18], the analysis has an objective to explore whether higher family support is associated with less risk of smoking behaviour, whilst controlling for demographic variables, focusing especially on gender, age, and country differences on the impact of family support on adolescent smoking. In line with this objective, the first hypothesis was that higher parental support is associated with a lower adolescent smoking risk. The second specific objective was to identify whether in having higher family support, adolescents can more easily pass the challenges during their transitions from early to late adolescence. Noting that the strength of the relationship between family support and adolescent smoking varied across countries, a second hypothesis was formulated. It claims that differences in smoking prevalence between 11- and 15-year-olds (e.g., during a period of transition from early to late adolescence) in each country appear to be related to the strength of the relationship between family support and adolescent smoking. 

## 2. Materials and Methods

### 2.1. Participants and Data Collection

The current study utilized data collected in 2017/2018 in the frame of the cross-national Health Behaviour in School Aged Children (HBSC) survey with support from the World Health Organization (WHO, Europe) [28]. It was completed in 44 European countries and regions (considered alone as countries, i.e., England, Scotland, and Wales), and Canada. More detailed background information about the study is provided on its website [28], in the international report [29,30], and research protocol [31]. 

The population selected for sampling included 11-, 13-, and 15-year-old adolescents. Sampling was conducted in accordance with the structure of national education systems within countries. In the majority of countries, the primary sampling unit was the school class, and students of the 5th, 7th, and 9th grades were targeted. School response rates within countries were in the range from 15.6% (Germany) to 100% (Bulgaria and Kazakhstan) (median 82.9%) [30].

The data were collected by means of self-report standardized questionnaires. They were gathered on young people’s health and wellbeing, and healthy and risky behaviours, and social context of young people life [31]. The surveys were administrated in school classrooms. Researchers strictly followed the standardized international research protocol to ensure consistency in survey instruments, data collection, and processing procedures [31,32]. Student response rates within participating classes varied between 42.0% (Sweden) and 98.6% (Albania) (median 83.2%) [30]. National datasets were cleaned by HBSC data managers and merged into the international dataset.

### 2.2. Ethics

The study was conformed to the principles for research outlined in the World Medical Association Declaration of Helsinki involving health promotion and safeguard, well-being, and rights of human subjects. National teams obtained ethical consent from the institutional ethics committee(s), when required. Parental consent was passive in most countries. Pupils were informed orally and in writing that participation in HBSC was voluntary. Students did not provide any personal details (such as name, classroom, teacher), making them completely anonymous and ensured the students’ confidentiality [31]. 

### 2.3. Measures

Current cigarette smoking was the main dependent variable of this study. It was assessed by the following question: ‘On how many days (if any) have you smoked cigarettes in the last 30 days?’ with the response options: 1 = never; 2 = 1–2 days; 3 = 3–5 days; 4 = 6–9 days; 5 = 10–19 days; 6 = 20–29 days; and 7 = 30 days (or more). In analyses, non-smokers (the first response option) were compared with those who reported smoking at least 1–2 days (all remaining response options). 

*Family support.* The main independent variable of adolescent smoking was family support. The survey question explored adolescents’ perceptions of how much supported they felt by their families. Family support was measured using a Family dimension (4 items) of the Multidimensional Scale of Perceived Social Support (MSPSS) [33]. Young people were asked how they feel about the following statements: *My family really tries to help me; I get the emotional help I need from my family; I can talk about problems with my family; My family is willing to help me make decisions.* The respondents rated each item on a seven-point Likert-type scale, ranging from “*very strongly disagree*” (0 score) through to “*very strongly agree*” (6 scores). The sum score was calculated as a sum of response scores to the four questions on family support ranging from 0 to 24 points (higher score corresponded to higher family support). Following previous studies [34,35], sum score 20 or more on MSPSS was categorised as high perceived family support. Cronbach’s alpha of the scale was 0.937.

*Controlled variables.* The analysis models were controlled for the effect of gender, age, family structure, and family affluence.

*Family structure.* The family structure variable examines with whom an adolescent lives all or most of the time, including biological mother and father, stepmother (father’s partner), stepfather (mother’s partner), living in foster or children’s home, or living with someone/somewhere else. Within the present analysis, the categories that were created comprise the groups of adolescents who live with both biological parents, and all the others. 

*Family affluence* was assessed through the Family Affluence Scale (third revision), which was specially developed for the HBSC study [36]. The scale is a validated measure for material affluence of household based on the following six items owned by the family: number of computers, number of cars, number of bathrooms, number of travels/holidays abroad, having own bedroom, and having a dishwasher. A family affluence score (FAS) was calculated by summing the points of the responses to these six items. Higher FAS values indicated higher family affluence. In accordance with the HBSC reports [18,29], this indicator was recoded into country-specific three groups. The first group included those in the lowest 20% (reference group), the second included those in the medium 60%, and the third group included those in the highest 20% of the FAS. 

### 2.4. Statistical Analysis

Two data files were used in analysis. The first file included 195,966 individual records from 42 countries. Data from three more countries (Azerbaijan, Greenland, and Norway) remained unused due to different methodologies for assessing cigarette smoking. The second file (Appendix A) included 42 records, which represented aggregated data by 42 countries, and grand totals, which were estimated from individual records weighting them by the proportion of respondents in each country. 

The effect of family support on cigarette smoking was assessed using Poisson regression with robust variance estimates [37,38,39], as an alternative model to the linear and logistic models [40]. In this model, the strength of association between family support and cigarette smoking was expressed as the prevalence ratio (PR), which meant the ratio between smoking prevalence among adolescents with low family support and smoking prevalence among adolescents with high family support. Regression analysis was performed separately in groups of boys and girls adjusting data for age, FAS, and family structure. Effects of interactions ‘family support × FAS’ and ‘family support × family structure’ were tested. The model goodness-of-fit to existing data was controlled with the Pearson χ^2^/df statistic (values 0.8 to 1.2 indicated good model fit to existing data [39]). Using country-level estimations, the difference in smoking prevalence between 11 and 15 years of age was calculated, and it was related both to the mean sum score of family support and to the effect size of family support while the last two estimates were calculated using data of 11-, 13-, and 15-year-olds. The strength of relationship between estimates was assessed using Pearson correlation coefficient *r*. As can be seen, this study analysed individual-level and country-level factors, but we did not apply multilevel models due to their limited efficiency for the second objective of the current research.

The analyses were performed using the SPSS statistical package (version 21; IBM SPSS Inc., Chicago, IL, USA, 2012). All reported *p*-values were from two-sided statistical tests and *p*-values ≤ 0.05 were considered statistically significant. 

## 3. Results

### 3.1. Current Cigarette Smoking 

Overall, prevalence of current cigarette smoking was similar in boys and girls (Table 1). Between the ages of 11 and 15 years, levels of smoking increased from approximately 2% of boys and 1% of girls to 16% of boys and 15% of girls, thus, the increase (14 percentage points) was approximately equal among both boys and girls. The prevalence of smoking varied greatly across countries (Appendix A). For example, at age 15, the prevalence of smoking in boys ranged from less than 7% in Kazakhstan and Iceland to 31.6% in Lithuania, and in girls this prevalence ranged from less than 3% in Armenia and Kazakhstan to 38% in Bulgaria. The largest age difference in current smoking prevalence among 11- and 15-year-olds was observed among Lithuanian boys (26 percentage points) and among Italian girls (32 percentage points). 

### 3.2. Family Support

Over two thirds of adolescents reported high levels of support from their family (72% of boys and 70% of girls). Family support was found significantly decreased with age in most of the countries, with the boys reported high level of family support more often than girls (Table 2). Wide cross-national variation was observed, with prevalence ranging from 30% among 15-year-old boys in Bulgaria to 94% of 11-year-old girls in Albania and North Macedonia (Appendix A).

Family support was associated with social factors of the family. Namely, there was seen a decrease in proportion of high family support in low affluence families in most of the countries. The adolescents from non-intact families were less likely to report having high family support compared to their peers from intact families (Table 3).

### 3.3. Association between Family Support and Smoking

The adolescents who reported having low family support also were more likely to smoke compared to their peers who reported having high family support. The strength of association between the reduction in family support and increase in smoking prevalence was greater among girls compared to boys and decreased by age for both boys and girls (Table 4). 

In almost all countries those adolescents who have low family support have higher smoking prevalence than their peers with high family support. The magnitude of effects that were estimated with PRs varied between 1.25 (Germany) and 4.85 (Malta) among boys and between 1.27 (Bulgaria) and 4.72 (North Macedonia) among girls (Appendix A). 

In the majority of countries, smoking prevalence was significantly affected by family structure as adolescents living in non-intact families had higher likelihood of smoking than those living in intact families. In cumulative data from 42 countries, among boys aged 11–15-year PR = 1.51 (95%CI: 1.43–1.60), and among girls of the same age PR = 1.56 (1.48–1.65). An interaction between family structure and family support was tested. Among boys, no significant interaction between these variables was found, while among girls, family support had stronger protective effect on the risk of smoking among girls living with both parents (in cumulative data, PR = 2.35 (2.21–2.51), *p* < 0.001) than among girls living in non-intact families (PR = 1.96 (1.80–2.20), *p* < 0.001). This interaction was significant in five countries (Albania, Denmark, Hungary, Luxembourg, and Scotland). Effects of family affluence on current smoking were not explicitly stated, either by the countries nor in the cumulative data. The component of interaction between family affluence and family support was not significant between adolescents of both sexes.

### 3.4. Association between Family Support and Smoking Prevalence, by Aggregated Data of 42 Countries 

Further analysis was performed using aggregated data from 42 countries (Appendix A). Table 5 shows Pearson coefficients of correlation between the prevalence of current cigarette smoking and the percentage of adolescents who reported high level of family support in 42 countries, by gender and age. Overall, among 11–15-year-olds, no significant correlation between the selected estimations was found, either among the boys nor among the girls. Considering the gender and age of the subjects, a significant correlation was found in the youngest groups of adolescents (in 11-year-old boys and 11–13-year-old girls). A negative correlation sign shows that in countries with greater percentage of adolescents who reported high level of family support it could be expected a lower smoking prevalence. However, the strength of correlation diminished by adolescent age, remaining slightly stronger among girls than among boys.

Table 6 shows Pearson coefficients of correlation between the prevalence of current cigarette smoking and the strength of the association between family support and adolescent smoking that were estimated using aggregated data from 42 countries. A highly significant relationship was found between the selected variables when analysing the data of 13- and 15-year-old boys and girls. A significant relationship was also confirmed in the joint 11–15-year-old teen sample. These data suggest that countries with a stronger effect of family support in reducing smoking risk have indicated lower rates of adolescent smoking prevalence.

### 3.5. Association between Family Support and Smoking Transitions in Adolescence

Analysis of aggregate data at the population level highlighted differences in smoking prevalence between 42 countries and revealed a variability in terms of the prevalence jump from 11 to 15 years, e.g., during a period of transition from early to late adolescence (see Section 3.1). We tested whether this change correlated with family support. The latter variable was also estimated at the population level of the country using data from all three age groups (11-, 13-, and 15-year-olds).

The scatter diagrams in the Figure 1 indicate no significant relationship between the increase in current cigarette smoking prevalence from 11 to 15 years and the percentage of respondents who reported high family support among neither boys nor girls. Pearson coefficients of correlation for these associations were 0.234 (*p* = 0.136) and –0.146 (*p* = 0.356), respectively, among boys and girls. Correlation analysis, meanwhile, revealed that the relationship between the increase in current cigarette smoking prevalence from 11 to 15 years, and the strength of the association between family support and adolescent smoking is highly significant (Figure 2). For this relationship, the Pearson coefficients of correlation were negative among both boys (–0.443; *p* = 0.003) and girls (–0.440; *p* = 0.004). This finding suggests that countries with a stronger effect of family support in reducing smoking risk have indicated a lower increase in the current cigarette smoking prevalence during the age period from 11 to 15 years.

## 4. Discussion 

The current study examined similarities and differences in perceived family support and its impact on overcoming the risk of smoking during adolescents’ transition from early to late adolescence. Data of the survey among 11–15-year-old adolescents from 42 countries that participated in the cross-national HBSC study in 2017/2018 were employed [29]. The results of analysis confirmed both hypotheses that were raised for the objectives of the study. First, using the individual data records, it was found that higher family support is associated with a lower adolescent smoking risk. Second, using aggregated country data, it was revealed that the countries with a stronger mean effect of family support in reducing smoking risk have indicated lower rates of adolescent smoking prevalence as well as a lower increase in the current cigarette smoking prevalence during the age period from 11 to 15 years. In summary, these findings allow us to conclude that family support is an important asset helping adolescents to overcome the risk of smoking traversing adolescence. 

The findings suggest that most young people feel family, generally their parents, supporting them. Proportion of adolescents who reported high family support in this HBSC wave remained almost the same as in the previous HBSC wave in 2013/2014 [18] (pp. 31–34). In line with previous research [41], our results indicate that perceived family support decreases with age. Traditionally older adolescents report having less family support compared to their younger counterparts. This result clearly reflects the beginning of the individualization process where the relationships with family members move from asymmetrical to a more symmetrical interaction being the adolescents treated as more autonomous individuals [4]. According to our findings, boys reported having higher level of family support compared to girls, which is in line with recently manifested research that girls, more often than boys, start problems of connectedness with parents in early adolescence, at the age of eleven [42]. Current research considers the parents–boys relationship to be more centred on independence and a higher need of psychological separation, while the parents–girls relationship is simultaneously based on independence but also connectedness, intimacy and reciprocity at the same time [43]. 

The associations of adolescent smoking behaviour with familial or parental variables have been extensively examined [44,45,46,47]. Although various variables to describe family functioning and parenting have received a great deal of attention [15,48], in this study we focussed on the associations of current smoking with family support. Only a few articles were found in the literature to analyse such an association [18] (pp. 31–34). Moreover, parenting includes several dimensions, including support and control. Parental support has been described as the variation in the amount of parental responsiveness and warmth, such as responding to the child needs, while parental control is a continuum that ranges from restrictiveness to permissiveness [49]. The effects of parental support and parental control on early adolescence smoking was analysed in a longitudinal study in the Netherlands [50]. Logistic regressions demonstrated that low parental control predicted adolescent smoking initiation but neither support nor control predicted adolescent smoking increase or continuation. Parental smoking status was important in adolescent smoking continuation and cessation. Unfortunately, it is beyond the scope of the present study to further analyse different components of family support. 

The results of the study showed a significant association between family support and smoking prevalence, indicating a positive effect of family support on prevention of smoking among adolescents. This finding was in line with the previous research supporting the idea that family support can help to form the most important basic values, attitudes and patterns of behaviour making adolescent transitions easier [51,52,53]. Consequently, family support has an overall protective effect on the risk of smoking among 11–15 years old adolescents. However, the strength of the association and the protective effect may vary, depending on other family characteristics, such as the material status of the family, parental monitoring and control, parental communication, and parenting styles. Research conducted by Mahabee-Gittens et al. (2012) revealed that higher parental monitoring and the attitudes towards smoking are significantly associated with recent smoking and ever smoking among US adolescents [54]. Aho et al. (2018) also found a significant protective effect of parental involvement (tested as a composite measure of parent–child relationship, family connectedness, and parental monitoring) on Finish adolescents’ risk of smoking [55]. Research conducted by Moore and Littlecott (2015) shows that higher family SES was associated with significantly lower likelihood of smoking and other health risk behaviours among Welsh adolescents [56]. These findings suggest that family socioeconomic status and family support may have independent and combined effect on young people’s risk behaviours, particularly on the risk of smoking, while family support may have stronger protective effect on the risk of smoking among adolescent from lower affluent families [56]. We did not find such associations in the current research. Instead, we found that family structure or living with both biological parents is a more important factor than family affluence. This factor may have an interaction with other familial factors in prevention of adolescent smoking [44]. These combined effects of family support and other determinants of family functioning could explain the non-significant association between the increase in current cigarette smoking prevalence from 11 to 15 years and the percentage of boys and girls who reported high family support. 

This study is exceptional because it involved adolescents from 42 countries. This allowed the analysis of variable associations at the country-level. One such analysis revealed that the prevalence of adolescent smoking was lower in countries where the greater proportion of adolescents felt high family support. This association was slightly stronger among girls than among boys, but it was significant only among the youngest adolescent groups. The prevalence of adolescent smoking had also a negative correlation with the strength of association between family support and adolescent smoking at country-level; however, in contrast with the previous association, it was significant in older adolescent groups only. Another country-level analysis found a significant correlation between the increase in adolescent smoking prevalence from 11 to 15 years and the strength of association between family support and adolescent smoking at country-level. The analysis of age-related developmental trajectory in adolescent health behaviours is important as such trajectories may also track into adulthood [57,58].

These results imply that the age-related increase in smoking prevalence during adolescence is less pronounced in countries where family support can be regarded as stronger protective factor against smoking than in countries where its impact is weaker. These findings are in line with previous research indicating the important role of family support during adolescents’ transition from early to late adolescence, especially as close family relationships can ameliorate the impact that adversity has on lifespan physical health [16]. Moreover, a high level of perceived family support is related to lower levels of risk behaviours reducing their risk behaviours [21,25] and it is a protective factor for children in adverse environments [24]. The mentioned increase in smoking prevalence at country level was not significantly correlated with the proportion of adolescents who reported high family support. The findings from examination of these associations may be generalized to support an epidemiologically based inference that the preventive effect or delay in the onset of many life-threatening conditions in the country does not depend on the extent of preventive measures but depends on their effectiveness. This means that the relationship between family support and smoking at the country level was moderated by the strength at the country-level of association between family support and adolescent smoking. Due to the unique cross-national design of our study, we did not find confirmation of this assumption among other studies. However, a preliminary analysis of HBSC data shows that this assumption is also valid for other outcome variables (e.g., alcohol consumption, and low life satisfaction), as well as observed in previous HBSC waves (e.g., in 2013/2014). Thus, the observed regularity deserves further investigation. Family can have important protective and preventive role, but also detrimental, such as being a supplier of alcohol to under-aged people [59]. The study implies that the well-documented age-related increase in smoking prevalence during adolescence is less pronounced in countries where family support can be regarded as stronger protective factor against this behaviour than in countries where its impact is weaker.

### Strengths and Limitations

The study is the product of an international network of researchers who work in topic-focused groups that collaborate to researching adolescent health. The research protocol includes scientific rational, international mandatory questions, and required procedures for sampling, data collection, and preparation of data set for ensuring high quality data. The measures of family life were based on valid scales. The use of large, nationally representative sample and the inclusion of 42 countries increases the generalizability of our finding. The analytical procedure facilitates the analysis of the relationships between family support and multiple self-reported aspects of adolescence transformation. 

There are several limitations to this research. The measure of outcome variable in this study was cigarette smoking in the past 30 days. This may misclassify some respondents who smoked cigarettes occasionally but did not smoke in the past 30 days or used e-cigarettes to smoke. There is also a likelihood of recall bias with a question covering the past 30 days; such a time frame applies to many population-based studies of youth lifestyles, so the bias would be consistent across studies. Moreover, using sensitive questions can also be affected by the possibility for social fear bias in adolescent responses. However, every effort was made to minimize that possibility by ensuring strict anonymity of respondents. This study relied only on self-reported data, although these data are considered to hold the most valid information when studying subjective measures such as relations with parents. The proportion of current cigarette smoking among 11-year-old adolescents, especially among girls, was relatively small (1–2%); therefore, the estimations of associations should be considered with caution. This study did not measure peer influence that may play significant role in preventing or promoting adolescent smoking [60]. Cultural factors may also contribute to family supports. Instead, we relied on prior studies’ findings of these cultural contexts [61,62,63]. Future studies should attempt to study country-level factors (e.g., the tobacco control legislation and policy) that contribute to cultural differences in the association between family support and development of behaviour in adolescence transition. Our study was cross-sectional in nature; therefore, the findings of such a study can only suggest associations but not causation [64]. Finally, we assessed the change in smoking prevalence during adolescence by comparing reports of 11- and 15-year-old adolescents in a cross-sectional survey, and further analysis of the associations were performed at the country level. Future research should continue to study the long-term associations between individual adolescent transitions such as family support and health behaviour trajectories. Particular attention should be paid to adolescents who are just trying to smoke (smoke 1–2 cigarettes per month) as they are most in need of family help and can benefit more than regular smokers.

## 5. Conclusions 

This study is among the first to examine cross-national similarities and differences in perceived family support among adolescents and its impact on overcoming the risk of smoking during their transition from early to late adolescence using data from many countries. The findings indicate that family support during adolescence exerts a persistent influence on diminishing risk of adolescent smoking. The results also imply that the age-related increase in smoking prevalence during adolescence is less pronounced in countries where family support can be regarded as a stronger protective factor against smoking than in countries where its effect is weaker. These results show that family support is a critical component to be incorporated in prevention and intervention programs for adolescent smoking.

### Implications of the Study

The present study reveals that high family support has a protective effect on the risk of smoking during the early phases of adolescents. The supportive family environment can alleviate the negative influence of other adverse conditions in the family, such as lower material status, lower monitoring on behalf of the parents, difficult communication with parents, etc. Development of specialized services for parent counselling as well as improvement of the communication between parents, teachers, and adolescents thorough special programs and school-based activities may strengthen the parental involvement and skills and reduce the prevalence of smoking among young people.

## Figures and Tables

**Figure 1 ijerph-18-12739-f001:**
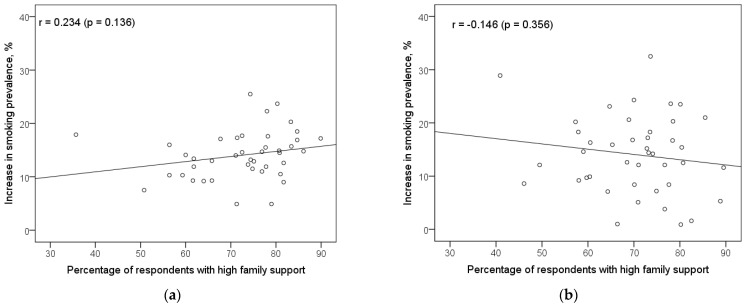
Scatter diagrams of the association between the increase in current cigarette smoking prevalence from 11 to 15 years of age and the percentage of adolescents who reported high family support, by gender, (**a**) Boys, (**b**) Girls.

**Figure 2 ijerph-18-12739-f002:**
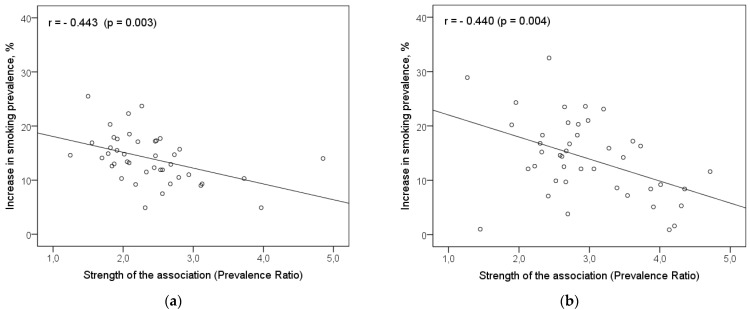
Scatter diagrams of the association between the increase in current cigarette smoking prevalence from 11 to 15 years of age and the strength of the association (Prevalence Ratio) between family support and adolescent smoking, by gender, (**a**) Boys, (**b**) Girls.

**Table 1 ijerph-18-12739-t001:** Summary data on smoking prevalence in adolescents from 42 countries, by gender and age.

Age	Proportion (%) of Adolescents Who Reported Current Cigarette Smoking ^1^			
	Boys		Girls	
	Proportion	SD	Proportion	SD
11 years	1.84	1.45	1.09	1.42
13 years	5.00	2.37	4.59	2.63
15 years	15.88	5.12	15.15	7.82
Increase in the smoking prevalence from 11 to 15 years of age	14.04	4.47	14.06	7.32

^1^ Data were weighted by country sample size. SD: standard deviation of the proportion estimates in 42 countries.

**Table 2 ijerph-18-12739-t002:** Summary data on high level of family support in adolescents from 42 countries, by gender and age.

Age	Proportion (%) of Adolescents Who Reported High Level of Family Support ^1^			
	Boys		Girls	
	Proportion	SD	Proportion	SD
11 years	78.90	10.03	78.77	10.72
13 years	72.36	11.45	69.09	11.77
15 years	65.76	12.80	62.72	11.72
11–15years	72.41	10.92	70.19	10.83

^1^ Data were weighted by country sample size. SD: standard deviation of the proportion estimates in 42 countries.

**Table 3 ijerph-18-12739-t003:** Summary data on the relationship of high level of family support with family affluence and family structure in adolescents from 42 countries, by gender and age.

Characteristics of the Family		Proportion (%) of Adolescents Who Reported High Level of Family Support ^1^	
		Boys	Girls
Family affluence	Low	65.6	63.1
	Medium	71.6	68.8
	High	74.2	72.4
	*p*	<0.001	<0.001
Family structure	Intact family	73.5	71.7
	Not intact family	62.5	58.1
	*p*	<0.001	<0.001

^1^ Data were weighted by country sample size. *p* was estimated using the χ^2^ test.

**Table 4 ijerph-18-12739-t004:** Summary data of the effect of family support on current cigarette smoking in adolescents from 42 countries, by gender and age.

Age	Estimates of Prevalence Ratio ^1^					
	Boys			Girls		
	PR	(95% CI)	*p*	PR	(95% CI)	*p*
11 years	2.93	(2.43–3.54)	<0.001	4.00	(3.17–5.02)	<0.001
13 years	2.39	(2.14–2.66)	<0.001	3.16	(2.82–3.54)	<0.001
15 years	1.57	(1.47–1.67)	<0.001	1.89	(1.77–2.01)	<0.001
11–15 years	1.81	(1.71–1.91)	<0.001	2.19	(2.08–2.31)	<0.001

^1^ Data were weighted by country sample size and adjusted for family affluence, family structure, and age in 11–15-year-olds. PR: prevalence ratio; CI: confidence interval.

**Table 5 ijerph-18-12739-t005:** Correlation between the prevalence of current cigarette smoking and percentage of adolescents who reported high level of family support in 42 countries, by gender and age.

Age	Pearson Coefficients of Correlation ^1^			
	Boys		Girls	
	*r*	*p*	*r*	*p*
11 years	–0.447	0.003	–0.462	0.002
13 years	–0.251	0.109	–0.397	0.009
15 years	0.140	0.377	–0.186	0.237
11–15 years	–0.036	0.820	–0.245	0.118

^1^ All countries were considered having equal weights.

**Table 6 ijerph-18-12739-t006:** Correlation between the prevalence of current cigarette smoking and the strength of the association between family support and adolescent smoking in 42 countries, by gender and age.

Age	Pearson Coefficient of Correlation ^1^			
	Boys		Girls	
	*r*	*p*	*r*	*p*
11 years	–0.072	0.686	–0.221	0.230
13 years	–0.335	0.030	–0.310	0.046
15 years	–0.440	0.004	–0.539	<0.001
11–15 years	–0.474	0.002	–0.515	<0.001

^1^ All countries were considered having equal weights.

## Data Availability

The data presented in this study are available on reasonable request from the HBSC Data Management Centre, University of Bergen, Norway (dmc@hbsc.org).

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
