# Peer review of "Family Support as Smoking Prevention during Transition from Early to Late Adolescence: A Study in 42 Countries"

_ijerph, 2021, doi:10.3390/ijerph182312739_

Round 1

Reviewer 1 Report

This cross-national study was to explore whether family support is associated with risk of smoking during the transition from early (11 years) to late (15 years) adolescence. The study has potential to be informative and the methods were appropriate and properly conducted. However, several questions remain in this study:

  1. How to measure the FAS and how is this indicator divided into low, medium and high?
  2. I suggest using the multilevel linear models to model of hierarchical or nested data structures. The level 1-units are individual students; the level 2-units are the 42 countries. In addition, the country-level GDP per capital could be included in the model for controlling.
  3. The tobacco control legislation and policy were related to tobacco use in adolescents. How to deal with the possible problems of different policies in 42 countries?

Author Response

Dear Reviewer,

Thank you for your thorough analysis of our article, its overall positive evaluation, constructive comments and suggestions. We explained point-by-point the details of the revisions in the manuscript as described below in italic. Changes in the manuscript text are with "Track changes" function in Microsoft Word, so that changes are easily visible (see attached document). As there were many changes in the manuscript, we uploaded the final draft in pdf version without "Track changes" in text. The numbers of lines were taken from this document.

General comment: This cross-national study was to explore whether family support is associated with risk of smoking during the transition from early (11 years) to late (15 years) adolescence. The study has potential to be informative and the methods were appropriate and properly conducted. However, several questions remain in this study:

Comment 1: How to measure the FAS and how is this indicator divided into low, medium and high?

Response: A separate paragraph (lines 161170) to explain the FAS has been added:

"Family affluence was assessed through the Family Affluence Scale (third revision), which was specially developed for the HBSC study [37]. The scale is a validated measure for material affluence of household based on the following six items owned by the family: number of computers, number of cars, number of bathrooms, number of travels/holidays abroad, having own bedroom, and having a dishwasher. A family affluence score (FAS) was calculated by summing the points of the responses to these six items. Higher FAS values indicated higher family affluence. In accordance with the HBSC reports [29,30], this indicator was recoded into country-specific three groups. The first group included those in the lowest 20% (reference group), the second included those in the medium 60%, and the third group included those in the highest 20% of the FAS."

By the way, the family structure was also explained in more detail (lines 155160):

"Family structure. The family structure variable examines with whom an adolescent lives all or most of the time, including biological mother and father, stepmother (father’s partner), stepfather (mother’s partner), living in foster or children’s home, or living with someone/somewhere else. Within the present analysis the categories that were created, comprise the groups of adolescents who live with both biological parents, and all the others."

Comment 2: I suggest using the multilevel linear models to model of hierarchical or nested data structures. The level 1-units are individual students; the level 2-units are the 42 countries. In addition, the country-level GDP per capital could be included in the model for controlling.

Response: Multilevel modelling is often used in the analysis of HBSC data to determine the prevalence of lifestyle factors and its relationship to individual factors (first-level factors) and country factors (second-level factors). Such an analysis could be applied in this study as well, but it would only be suitable for solving the first objective and testing the first hypothesis. The results on current smoking prevalence and strength between smoking and family support by countries (see supplement file for the article) are good illustration for the possibility of multilevel analysis. Unfortunately, the multilevel modelling method cannot be applied to solve the second objective and test the second hypothesis. The variables used in this case are complex and derived from the first objective. A dependent variable is "differences in smoking prevalence between 11- and 15-year-olds (e.g. during a period of transition from early to late adolescence) in each country" and one of independent variables is "the strength of the relationship between family support and adolescent smoking". All these variables are second-level factors. Of course, other country-level factors can be added to this linear relationship. We think that variables on the tobacco control legislation and policy, as you suggest in the third comment, would be more appropriate in this case than the GDP per capita.

In the light of this comment, section 2.4. Statistical Analysis was supplemented to explain why the multilevel analysis method was not used in the current study as follows (lines 192194): "As can be seen, this study analyzes individual-level and country-level factors, but we did not apply multilevel models due to their limited efficiency in the second objective of the current research."

Comment 3: The tobacco control legislation and policy were related to tobacco use in adolescents. How to deal with the possible problems of different policies in 42 countries?

Response: Thank you for this valuable observation. There is no doubt that tobacco control legislation and policy have an impact on the prevention of the smoking habit developed in adolescence. Unfortunately, the study of this problem goes beyond the scope of the present research. Therefore, in the section of Strengths and Limitations, we have just indicated that this is a topic for future research (lines 450454).

Reviewer 2 Report

Dear author(s),

I would like to thank the journal for giving me the opportunity to review this paper. I do feel there is room for improvement, not least with regards to the methods and results sections, but due to the overall merit of the study I would suggest some revisions.

Some comments and suggestions:

- Smoking is outside my field of research so I feel I'm not qualified to comment on the theoretical foundation for that part, but I did think that the introduction overall is relevant and on point, not least with the general description of adolescent development with regards to family support.

- There is something odd with the sentence on lines 56-59 – would it work to change “…but this role may be” to “and this role may be”?

- Lines 110-118. Here I would like to see a bit more detail on the sampling and data collection procedure. That is, how many schools were approached and how many accepted to participatete, and what was the participation rate within schools and countries (it currently only says over 80% in most countries, but here it would be preferrable to provide the range as well as either the mean or median participation rate).

- Line 121 – Here the main principles of the Helsinki declaration should be clearly highlighted.

- Lines 128-133 – This is probably one of my main concerns. Did the author(s) ever test any other way of categorizing the smoking variable, for example, into more than two categories? To my mind there is a qualitative difference in smoking once a day versus once a month, and since it’s a cross-sectional study, the response option 1-2 days might even reflect someone trying out their first cigarette. With all the options 2-7 lumped together, the effect of family support on heavy smoking might unduly conflate any milder or even non-effects of family support on irregular smoking, or from irregular smoking to heavy smoking.  I would strongly suggest to test and report the findings of a dose-response model (for example with no-smokers vs infrequent smoking (1-2 times) vs regular smoking (the other options), if not as the main result then at least as a post-hoc follow-up of the main findings.

- Line 149 – more info is needed on both the items and the coding of the Family Affluence Scale (FAS)

- Line 157-159 -- I’m not familiar with the Poisson regression, so this might be self-evident for others, but to my mind it’s not clear why the author(s) used two separate sets of data (individual and aggregated) instead of a multilevel approach. Here a short explanation would be welcome with regards to the benefits of the current approach.

- Line 337-338 -- the author(s) note that family SES and family support may have independent and combined effects on young peoples risk behavior. In the current study, there are measures of both family affluence and family structure, in addition to the perceived support scale – however, I fail to see where the authors have tested for these potential interaction effects on smoking behavior. I would strongly suggest to test these interaction effects on the current data, and discuss the findings here in relation to previous research

- Line 371-373 – This sounds like a tautology, consider re-wording

Overall, while there is merit to the current version, the manuscript might be further improved by providing additional analyses. I would thus suggest the paper to be revised and resubmitted.

Best,
Reviewer

Round 2

Reviewer 1 Report

The manuscript has been corrected on reviewer' comments. No more comments.